# Potential Role of Biocontrol Agents for Sustainable Management of Fungal Pathogens Causing Canker and Fruit Rot of Pistachio in Italy

**DOI:** 10.3390/pathogens11080829

**Published:** 2022-07-25

**Authors:** Giorgio Gusella, Alessandro Vitale, Giancarlo Polizzi

**Affiliations:** Department of Agriculture, Food and Environment (Di3A), University of Catania, Via S. Sofia 100, 95123 Catania, Italy; giorgio.gusella@phd.unict.it (G.G.); gpolizzi@unict.it (G.P.)

**Keywords:** *Bacillus amyloliquefaciens*, biological control, canker management, fruit rots, pistachio diseases, *Trichoderma*

## Abstract

Pistachio (*Pistacia vera*) is an important Mediterranean crop. In Italy, pistachio is cultivated in the southern regions, of which Sicily is the main production area. Recently, the phytopathological situation of this crop has started to be updated, and new diseases have been discovered, studied, and reported. *Botryosphaeriaceae* spp. and *Leptosillia pistaciae* are major canker/rot pathogens, and *Cytospora pistaciae* and *Eutypa lata* have been reported as minor canker pathogens. In this paper, we evaluated different biological control agents, belonging to *Trichoderma asperellum, T. atroviride* and *T. harzianum*, as well as some *Bacillus amyloliquefaciens* strains, against above-mentioned pathogens. Results of dual culture assays showed that all the biological products, both fungi and bacteria, were able to inhibit the mycelial growth of the pathogens in vitro. Experiments using detached twigs showed no effect of biocontrol agents in reducing infections, except for *Neofusicoccum hellenicum* treated with *T. harzianum* T22 and *Leptosillia pistaciae* treated with *B. amyloliquefaciens* D747. Results of detached fruit experiments showed an efficacy ranging from 32.5 to 66.9% of all the biological products in reducing the lesions caused by *N. mediterraneum*. This study provides basic information for future research on biological control of pistachio diseases and future prospects for search of more effective biological control agents for canker diseases than those studied here.

## 1. Introduction

Pistachio (*Pistacia vera* L.) is an important Mediterranean crop, the production of which is mainly concentrated in Asia (70.8%) and Americas (26.5%), following the production in Europe (2%), Africa (0.5%) and Oceania (0.2%) [1]. In Italy, pistachio is cultivated in the southern regions, of which Sicily is the main production area. Catania province, and specifically the municipality of Bronte, is considered the traditional production area, whose orchards are defined “natural plantings”, since *P. vera* is grafted onto *P. terebinthus* (wild pistachio species) grown in the peculiar volcanic soil of the area with no rational design [2]. This represents a very important economic resource for Mediterranean basin, to the point that in 2010 the “Green Pistachio of Bronte” [Pistacchio Verde di Bronte] was officially registered as an Italian protected designation of origin (PDO) product [3]. However, mainly in Agrigento and Caltanissetta provinces, more recent pistachio orchards, commonly defined “new” orchards, are characterized by a rational design, irrigation, fertilization, and mechanical harvest [4]. Regarding diseases affecting this crop, only a few outdated studies were available until recently. Recent studies conducted in Italy updated the knowledge of fungal diseases of pistachio, specifically canker and dieback caused by *Cytospora pistaciae*, *Eutypa lata* and *Leptosillia pistaciae* (ex *Liberomyces pistaciae*) [5,6,7]. Among fruit and foliar diseases *Arthrinium xenocordella* and *Tuberculina persicina* were reported attacking fruit nuts [8,9] and molecular investigation confirmed the species *Septoria pistaciarum* as the main foliar pathogen of pistachio in Italy [10]. Moreover, field surveys conducted in the new orchards of Agrigento and Caltanissetta provinces revealed the presence of *Botryosphaeriaceae*, responsible for the destructive diseases named “Botryosphaeria Panicle and Shoot Blight” [11]. Species identified in Italy included: *Botryosphaeria dothidea*, *Neofusicoccum hellenicum* and *N. mediterraneum* infecting fruit clusters, leaves and shoots [11]. Among these, *N. mediterraneum* is the widest spread in the orchards. *Botryosphaeriaceae* species can infect many different hosts and because these species are not host specific, there is a high possibility to jump from one host to another, especially in the Mediterranean region where many different crops are cultivated one next to each other [12]. Management of Botryosphaeria canker and blight diseases is a crucial point for a grower in order to reach high yield; for this reason, fungicide applications in conjunction with other management practices, such as quarantines, monitoring of young plantings for any invasion by pistachio pathogens, regular inspections, accurate pruning, proper irrigation systems, cultivar evaluation and so on, represent crucial points [13]. 

The successful management of these diseases is certainly addressed not only in terms of reducing disease pressure in the field and/or postharvest, but especially when the control strategies are environmentally friendly and safe for workers and consumers. Following the European Commission’s “Farm to Fork Strategy” indications, a reduction of 50% in the usage of pesticides should be achieved by the year 2030, providing innovative and healthier alternatives than the ones presently used by growers and consumers. In this line, the restriction of many chemicals around the world has led to a new focus of the research on non-chemical/BCAs for controlling canker diseases [14]. In fact, the agrochemical industry has updated its priorities including the BCAs in research and programs for disease and pest control, both on pre-harvest and postharvest [15]. Many researchers are now focused on finding alternatives to synthetic pesticides, especially in the usage of microbial biocontrol agents (BCAs). Microbial biocontrol agents are usually fungal or bacterial strains derived from the phyllosphere, endosphere or rhizosphere able to manage many different plant pathogens through several mechanisms such as antibiosis, parasitism, competition, exclusion, induced resistance and growth enhancement [16]. Regarding fungal BCAs, almost 90% belong to different strains of *Trichoderma*, characterized as rapid colonizers, invasive, filamentous, opportunistic, avirulent and having a symbiotic relationship with plants [17]. Among the bacterial BCAs, those belonging to *Bacillus subtilis* species complex are widely investigated for their abilities in biocontrol. These bacteria have been isolated from many different ecological niches such as soil, plants, food, air and aquatic environments, showing direct (enhancing plant growth) and indirect (improving plant health) biocontrol mechanisms [18]. In Italy, since the phytopathological situation of pistachio has been updated in recent years, no studies on disease management were conducted. Growers are allowed to use few chemical pesticides, and concerns about their possible future restrictions certainly exist. For these reasons, the aims of this study were to conduct a screening on the potential of BCA trade products (antagonistic fungi and bacteria) based on in vitro dual culture assays and in vivo detached experiments against the main canker and fruit fungal pathogens of pistachio in Italy.

## 2. Results

### 2.1. In Vitro Dual Culture Assays

Since the assay × treatment interactions and treatment effects were significant (data not reported) the in vitro data regarding *Trichoderma*-fungal target dual culture were reported for each assay (Table 1). Based on these data, all *Trichoderma*-based formulates significantly inhibited the mycelial growth of tested fungal isolates, except for *L. pistaciae* in the first assay, for which no differences among treatments were detected, and for *T. harzianum* T22, which was revealed to be ineffective versus *E. lata* in the first assay. 

Based on the cumulative data, *Trichoderma atroviride* SC1 showed the best effects in reducing mycelial growth for *N. hellenicum*, *E. lata* and *L. pistaciae* (second assay) whereas all *Trichoderma* BCAs showed similar performances against *N. mediterraneum* and *B. dothidea* (Table 1). Moreover, fungal antagonists were able to overgrow on the pathogen’s colonies. Sometimes when the arrested growth of the two colonies in contact was observed, the antagonists started to overgrow on the pathogens over time (Figure 1).

Similar to previous dual-culture in vitro data, the assay × treatment interactions and treatment effects of *Bacillus*-based BCA formulates were also significant (data not reported), except for treatment effects against *L. pistaciae*, for which no differences were detected among considered *Bacillus* BCAs and treatment effects against *N. mediterraneum* in the first assay. Consequentially, data are shown for a single assay (Table 2). Based on these data, *B. amyloliquefaciens* QST713 and *B. amyloliquefaciens* D747 provided the best performances against *N. hellenicum* and *E. lata*, respectively. 

In addition, bacterial antagonists were able to induce a marked inhibition halos and to stop mycelial growth when in dual culture (Figure 2).

### 2.2. Detached Twig Assays

The in vivo performance evaluation of BCAs against fungal twig infections, effects of treatment and treatment × trial interactions were not significant versus disease parameter except only for the treatment effects on *N. hellenicum* and *L. pistaciae* infections, respectively (Table 3). Therefore, the two trials were combined, and average data presented in the Table 4.

The analysis of the data in terms of effectiveness ranking did not reveal any differences among the tested treatments for *N. mediterraneum*, *B. dothidea* and *E. lata*, while significant differences were detected for *N. hellenicum* and *L. pistaciae* (Table 4). In detail, only the bioformulation containing *T. harzianum* T22 was able to significantly reduce *N. hellenicum* twig infections when compared with the untreated control, whereas *B. amyloliquefaciens* D747 provided the best performances in reducing disease infections caused by *L. pistaciae* (Table 4). 

Sometimes, for the *Trichoderma* based formulations the growth and sporulation of the biocontrol agent on the woody tissues were clearly observed (Figure 3). 

Although no negative controls were included in this assay, previous experiments showed that healthy shoots (uninoculated and untreated) did not provide lesions.

### 2.3. Detached Fruit Assays

In the evaluation of BCAs against *N. mediterraneum* fruit infections, treatment effects were always significant, whereas treatment × trial interactions and trial effects were not significant both in terms of the average lesion diameter and decayed surface (Table 5). Consequentially, the data of the two trials were combined.

The analysis of the main effects of treatments on fruit decay is reported in Table 6. Based on these data, all BCA applications at the two tested dosages significantly reduced both decay lesion diameter and infected area, except for the bioformulations based on *B. amyloliquefaciens* MBI600 on diameter, which did not differ from untreated controls. Although the average control efficacy percentages (decay reductions relative to untreated control) ranged from about 32.5 to 66.9%, all *Trichoderma-* and *Bacillus*-based bioformulations did not always differ among them (Table 6, Figure 4).

## 3. Discussion

This study first shows data about the efficacy of biocontrol formulations against canker pathogens of pistachio in in vitro assays and experiments on detached twigs and fruits. Canker pathogens represent a serious threat for this crop, especially because of the difficulties in controlling them. Previous studies showed that the fungus *L. pistaciae* (described as *Liberomyces pistaciae*) is widespread among the traditional area of Bronte, representing an insidious limiting factor for the growers [6]. Recent investigations conducted in the new orchards of the island showed the presence of *Botryosphaeriaceae* species attacking pistachio [11]. Other studies also reported *C. pistaciae* and *E. lata* being sporadic among the orchards [5]. Most of the canker-pathogens, such as *Botryosphaeriaceae* spp. occur in complexes [19], and produce spores in flask-like structures, both asexual (pycnidia) or sexual (pseudothecia) embedded in the outer layers of the infected host tissue, making the penetration of the fungicides difficult [12,13]. Although many bioformulates have been tested among different key pathogens in pre-and postharvest, few studies are available regarding biological control of pistachio diseases. In our study, fungal biocontrol agents belonging to *Trichoderma* as well as *B. amyloliquefaciens* based formulations were tested in dual culture and in planta experiments. Dual culture assays revealed that all the biocontrol agents were able to inhibit the mycelial growth of the pathogens. In our assays it was observed that *Trichoderma* spp. grew over the pathogen colonies. It is well known that *Trichoderma* spp. grow quickly and compete for space, contributing to its ability to inhibit pathogens in dual-culture assays [20]. Further studies need to be conducted to reveal the endophytic colonization potential of these species as pruning wound protectants in the fields. For example, field studies conducted on almond affected by the band canker in California, showed promising results derived from the usage of Vintec^®^, resulted in 90 to 93% protection of pruning wounds in field trials [21]. Similarly, in the dual cultures with bacterial strains, a reduction of mycelial growth of the pathogens was observed, explained by the inhibition halos, probably due to the presence of antimicrobial compounds produced by the biocontrol agents.

In the detached twig assay the BCAs did not show an effective ability to contain the disease level, except for slight differences of *N. hellenicum* treated with *T. harzianum* T22 and *L. pistaciae* treated with *B. amyloliquefaciens* QST 713. The experiments of detached fruits were conducted inoculating *N. mediterraneum*. According to previous investigations in the new orchards, this species is widely distributed, commonly found associated with fruit black spots [11]. Findings of our experiments revealed that in our fruit detached experiments, BCAs provided a promising potential in decay reduction. Moreover, additional investigations need to be performed in order to ascertain the proper timing of applications and the potential effects on fruit quality. In California, interesting findings against *B. dothidea* (at that time considered the only agent of Botryosphaeria Panicle and Shoot Blight) were obtained testing the isolate CBCA-2 of *Paenibacillus lentimorbus* [22]. Chen et al. [22] demonstrated that this isolate caused a significant inhibition and malformation of pycnidiospores and hyphae, reducing the disease on detached leaves and protecting the pruning wounds before the inoculation with *B. dothidea*. We do not have many data regarding biological control of pistachio diseases. In Iran, promising results were obtained with the usage of *Streptomyces misionensis* to control *Paecilomyces formosus*, the causal agent of pistachio canker and dieback [23] and strains of *Streptomyces* and *B. subtilis* against *Phytophthora* spp. agent of pistachio gummosis [24,25,26]. The results of the present study indicate that BCAs can be a glimmer of hope in the management of pistachio canker diseases. Although canker diseases represent a serious concern for the growers, due the management difficulties, the complexity of etiological agents and the progression of the diseases in the field over the years, it should also be noticed that nut crops are highly susceptible to other important phytopathological issues, such as mycotoxin contaminations. 

Research focused on BCAs could also represent an important key point to prevent and control the toxigenic fungi in the field as well as in postharvest. The usage of BCAs for mycotoxin control could be part of a more holistic approach of disease management [15]. In fact, the usage of BCAs for disease management needs to be taken into consideration not only in terms of effectiveness against target pathogens but considering the whole ecosystem health. Biological control should improve environmental quality by diversifying the beneficial microorganism’s population in farmlands to avoid the occurrence and development of pathogens [27]. To this aim, the pistachio orchards with low disease pressure under suppressive natural conditions should be monitored to recover potential endophyte BCAs. In addition, biological control can generate multiple effects in food production, also affecting the economic development. The BCAs must provide an economic incentive to the end-users compared to other approaches of disease management [27]. This study shows first results of BCAs effectiveness, but further field trials are needed to ascertain the practical effectiveness of these products for the growers. Canker diseases are difficult to manage and require many different efforts to effectively control them. Although we know that certain group of fungicides effectively contain *Botryosphaeriaceae* infections [28], new management strategies, like the usage of BCAs, can be promising. Fungicide spray programs are not the only tool to control cankers diseases, but they should be implemented with many other practices, from the quarantines/inspections to the cultural methods [13]. Since pistachio production is increasing in Italy, it is crucial to identify new strategies and opportunities in terms of fruit quality and environmental sustainability. This study provided basic information for further research aimed to assessing the BCAs products under field conditions.

## 4. Materials and Methods

### 4.1. Fungal Pathogens and Biocontrol Agents

For the experiments conducted in this study, the following pistachio pathogens, stored in the fungal collection of the Department of Agriculture, Food and Environment (Dipartimento di Agricoltura, Alimentazione e Ambiente, Sezione di Patologia Vegetale, University of Catania), were selected: *B. dothidea* (isolate P89), *E. lata* (PV85), *L. pistaciae* (PV14 and PV30), *N. hellenicum* (P109), and *N. mediterraneum* (P107). All the fungal isolates were derived from previous published studies conducted in Italy and they were molecularly and morphologically characterized [5,6,11]. Biological products used in this study were fungal antagonists as well as bacterial ones (Table 7). Fungal antagonists used in this study were: *Trichoderma asperellum* strain T34 (T34 Biocontrol^®^, Biolchim, Medicina BO, Italy), *T. atroviride* strain SC1 (Vintec^®^, Belchim, Medicina BO, Italy), *T. harzianum* strain T-22 (Trianum-P^®^, Koppert, Monster, The Netherlands). Bacterial antagonists were *Bacillus amyloliquefaciens* ssp. *plantarum* strain D747 (Amylo-X^®^ LC, Biogard, Grassobbio, Italy), *B. amyloliquefaciens* strain MBI600 (Serifel^®^, BASF, Ludwigshafen, Germany), *B. amyloliquefaciens* strain QST 713 (Serenade ASO^®^, Bayer, Leverkusen, Germany).

### 4.2. In Vitro Dual Culture Assays

In vitro evaluation of the potential biocontrol agents against pistachio canker and fruit pathogens was assessed by the dual culture assay. For each experimental unit in the dual culture fungus-fungus, a mycelial plug of the pathogen and of the antagonist was cut with a cork borer (6 mm diameter) from a 5–8 days-old cultures grown on PDA and placed opposing one another on the outer edges (5 mm distant from the edges) of a 90-mm diameter PDA Petri dish. In the case of *B. dothidea*, *N. hellenicum*, and *N. mediterraneum*, considered fast growing fungi, the fungal BCAs were cultured 24 h before the pathogens, and the evaluation of their interaction was recorded 5 days after their culturing. In the case of *E. lata* and *L. pistaciae*, considered slow growing fungi, they were cultured 4 days before the antagonists and the results were recorded 7 days after their culturing. For the interaction evaluation, two rays (left and right) of the pathogen colony were measured. Macroscopic interactions were observed. Three Petri plates were used for each treatment, incubated at 25 °C, and the experiment was repeated once. Controls consisted of dual cultures of the same pathogen isolate cultured as described above. For the interaction with bacterial BCAs, the bacterial products were streaked out onto the PDA with a sterile needle eye, covering half of the Petri plate. The mycelial plug of the pathogen was placed on the opposite side (5 mm distant from the edge of the Petri plate). In the case of the interaction with fast growing fungi, the bacterial products were streaked 24 h before the pathogen, whereas in the case of *E. lata* and *L. pistaciae* (slow growing fungi), the bacterial products were streaked 2 days after the pathogens. The results were recorded 5 days after their culturing for fast growing fungi, whereas 9 days later for the slow growth pathogens. Two rays of the pathogen colony were measured (left and right) as well, and the distance between the pathogen and the bacterial colony line (inhibition halo) was also measured. For both experiments three Petri plate were used for each treatment, incubated at 25 °C, and the experiment was repeated once. Controls consisted of dual cultures of the same pathogen isolate cultured as described above. 

### 4.3. Detached Twig Assays

In this experiment, BCAs agents were tested in detached twigs. Specifically, a total of three 1-year-old shoots (~30 cm in length) of pistachio cv. “Bianca” were used for each pathogen and biocontrol agent interaction. Detached twigs (~ 30–35 cm in length) were disinfected by dipping them in a household bleach solution (3–3.5%) at 0.5% for 4 min and rinsing them twice in sterile deionized water (SDW). Twigs were left to dry on the laboratory bench overnight. Once completely dry, two wounds for each twig were made using a cork borer (6 mm diameter), with a distance as almost 10–15 cm in between. Each twig was immersed for 10 min in the BCAs water solution (the lowest label dosage according to the label recommendations) and air dried before applying the mycelial plug (6 mm diameter) of each pathogen onto the wounds. Each wound was sealed with Parafilm^®^ (Pechney Plastic Packaging Inc., Shelbyville, TN, USA) and the shoots were placed in a plastic container (60 × 40) with water at the bottom to create and keep high relative humidity and incubated in a growth chamber at 25 ± 1 °C with a 12 h photoperiod. Controls consisted of inoculation of untreated twigs. The experiment was repeated once. Length of the lesion were measured 3 weeks after the inoculations.

### 4.4. Detached Fruit Assays 

In this experiment, BCAs agents were tested on detached fruits against *N. mediterraneum*, the most spread and highly aggressive species found on fruit in Sicily [11]. A total of 6 pistachio fruits for each tested BCAs were collected in the field and brought to the laboratory. Fruits were surface disinfected dipping them in a 0.5% solution of household bleach (3–3.5%) for four minutes and rinsed in SDW. Fruits were left to airdry on the laboratory bench before treatment with the BCAs. Once completely dried, they were dip for 2 h in a water solution of BCAs at two different dosages, specifically low and medium label dosage. Controls consisted of fruits immersed in SDW (positive control) and inoculated with the pathogen. After 2 h of dipping, fruits were left to air dry for 15 h. Once dried, they were superficially wounded on the epicarp with an insulin needle and a 4 mm mycelial plug of *N. mediterraneum* was placed upside down on each fruit. Fruits were placed in plastic container and incubated in a growth chamber with a 12 h photoperiod at 23 °C. Average length of the lesions (two perpendicular axes or diameters) was measured after 4 days of incubation. Since the fruit spots were ellipsoidal in the shape the decayed area was further calculated according to the following formula:(1)$$\mathrm{Area}=\frac{D1}{2}\times \frac{D2}{2}\times \;\pi$$
where Area is the surface (cm^2^) infected by *N. mediterraneum*, D1 the major axis, D2 the minus axis and π = 3.14159….

The experiment was repeated once.

### 4.5. Data Analysis

Data about in vitro and in vivo performance of BCAs and fungicide from the repeated experiment were always analyzed by using the Statistica package software (version 10; Statsoft Inc., Tulsa, OK, USA). The arithmetic means of fungal mycelial growth, twig lesion and fruit decay variables were calculated, averaging the values of replicates for each treatment. Initial analyses of mycelial growth and disease amount values were performed by calculating F and *p* values associated to evaluate whether the effects of single factor (treatment) and treatment × assay (or trial) interactions were significant. In the post-hoc analyses, main treatment effects were evaluated and the mean values of fungal or disease parameters were subsequently separated by the Tukey’s honest significance difference test (α = 0.05).

## Figures and Tables

**Figure 1 pathogens-11-00829-f001:**
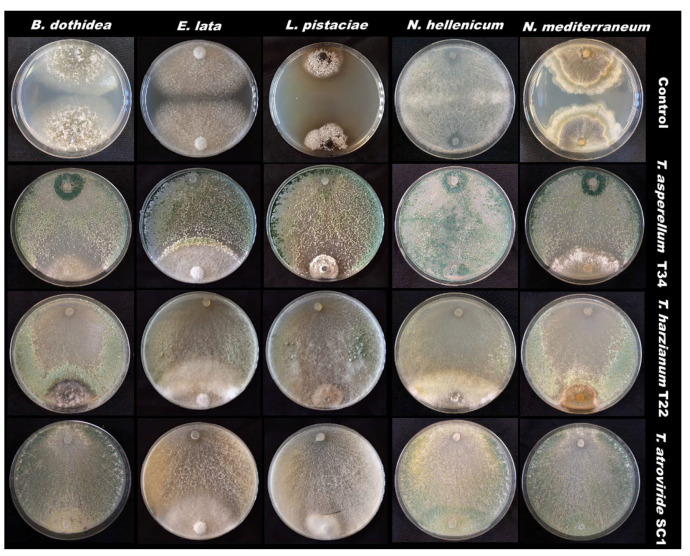
Dual culture assays with fungal antagonists.

**Figure 2 pathogens-11-00829-f002:**
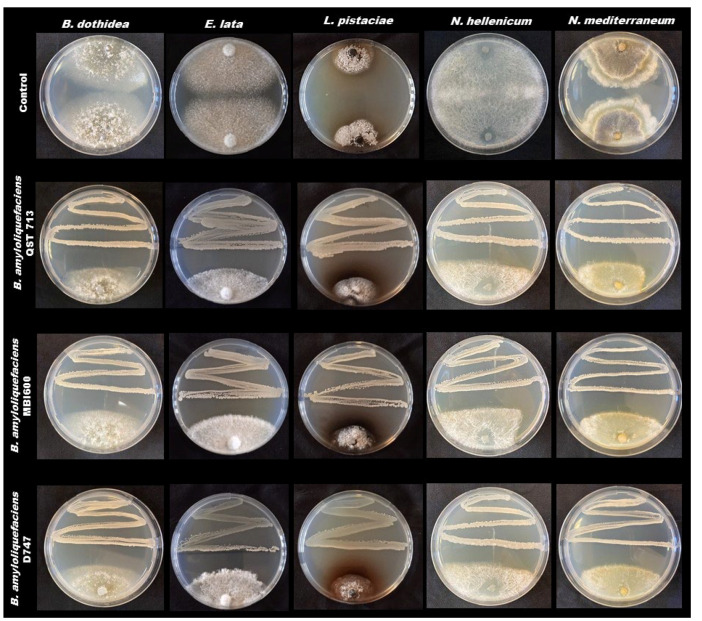
Dual culture assays with bacterial antagonists.

**Figure 3 pathogens-11-00829-f003:**
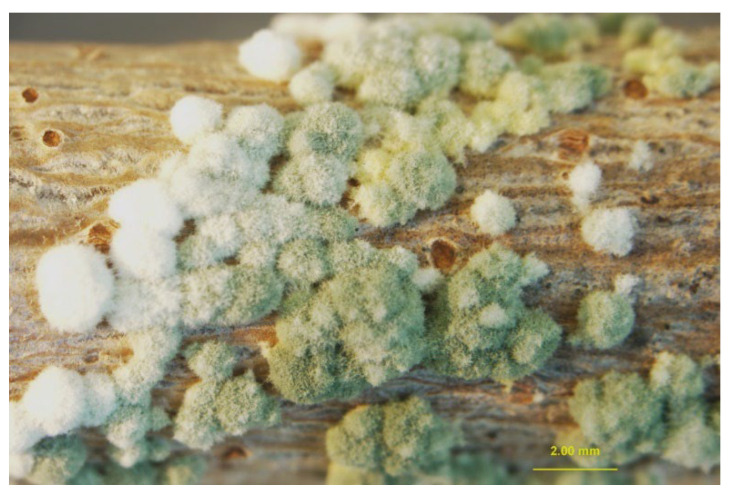
*Trichoderma asperellum* T34 sporulating on woody tissues of detached twig under high humidity levels at 25 °C for three weeks.

**Figure 4 pathogens-11-00829-f004:**
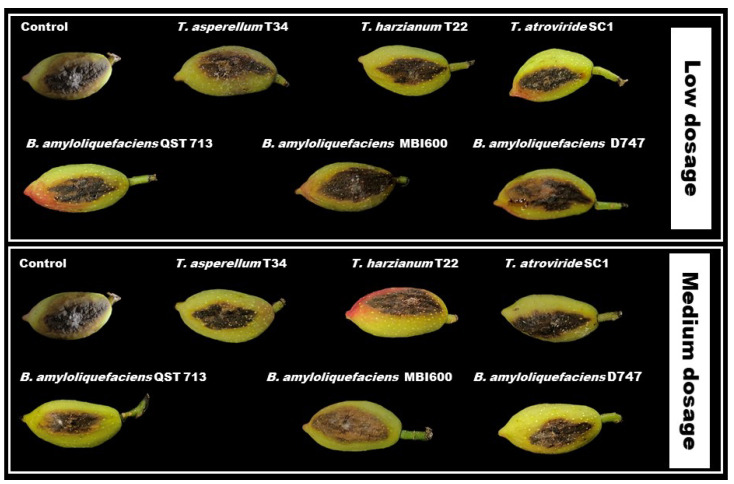
Lesion size on pistachio fruit (cv. Bianca) after inoculation with *Neofusicoccum mediterraneum* and treatment with low and medium dose of BCAs. Lesions were measured four days after inoculation and treatment.

**Table 1 pathogens-11-00829-t001:** In vitro effects of 3 antagonistic fungi in reducing mycelial diameter of Neofusicoccum mediterraneum, Neofusicoccum hellenicum, Botryosphaeria dothidea and Eutypa lata, Leptosillia pistaciae.

Treatment	First Assay ^y^				
	*N. mediterraneum*	*N. hellenicum*	*B. dothidea*	*E. lata*	*L. pistaciae*
Control	2.86 ± 0.06 a	4.04 ± 0.02 a	4.39 ± 0.14 a	3.84 ± 0.07 a	2.21 ± 0.02 ^ns^
*T. atroviride* SC1	1.45 ± 0.03 b	1.73 ± 0.14 b	1.52 ± 0.10 b	3.32 ± 0.09 b	2.00 ± 0.29
*T. asperellum* T34	1.37 ± 0.07 bc	1.57 ± 0.31 b	1.02 ± 0.12 b	3.28 ± 0.07 b	2.42 ± 0.22
*T. harzianum* T22	1.10 ± 0.07 c	2.08 ± 0.03 b	1.45 ± 0.10 b	3.72 ± 0.03 a	1.78 ± 0.03
	**Second Assay ^y^**				
	** *N. mediterraneum* **	** *N. hellenicum* **	** *B. dothidea* **	** *E. lata* **	** *L. pistaciae* **
Control	3.80 ± 0.10 a	4.53 ± 0.07 a	4.08 ± 0.01 a	4.84 ± 0.06 a	1.95 ± 0.01 a
*T. atroviride* SC1	1.27 ± 0.02 c	1.80 ± 0.05 c	1.42 ± 0.06 b	3.00 ± 0.00 c	1.22 ± 0.07 c
*T. asperellum* T34	2.03 ± 0.07 b	2.50 ± 0.10 b	1.38 ± 0.03 b	3.50 ± 0.08 b	1.38 ± 0.02 bc
*T. harzianum* T22	1.38 ± 0.04 c	2.78 ± 0.11 b	1.50 ± 0.06 b	3.57 ± 0.09 b	1.47 ± 0.07 b

^y^ Values derived from three replicated Petri dishes with standard error of the mean (SEM). Means followed by different letters within the column are significantly different according to Tukey’s honest significance difference test (*α* = 0.05). ns = not significant data.

**Table 2 pathogens-11-00829-t002:** In vitro effects of 3 antagonistic bacteria in reducing mycelial diameter of *Neofusicoccum mediterraneum*, *Neofusicoccum hellenicum*, *Botryosphaeria dothidea*, *Eutypa lata* and *Leptosillia pistaciae*.

Treatment	First Assay ^y^				
	*N. mediterraneum*	*N. hellenicum*	*B. dothidea*	*E. lata*	*L. pistaciae*
Control	2.60 ± 0.15 ^ns^	4.00 ± 0.00 a	4.17 ± 0.06 a	4.82 ± 0.12 a	2.38 ± 0.02 ^ns^
*B. amylo.* QST713	2.05 ± 0.19	3.20 ± 0.15 b	2.87 ± 0.12 b	3.07 ± 0.03 b	2.57 ± 0.07
*B. amylo.* MBI600	2.03 ± 0.02	3.48 ± 0.16 ab	2.68 ± 0.08 b	2.72 ± 0.07 bc	2.70 ± 0.05
*B. amylo.* D747	2.05 ± 0.13	3.50 ± 0.06 ab	2.85 ± 0.20 b	2.58 ± 0.16 c	2.43 ± 0.23
	**Second Assay ^y^**				
	** *N. mediterraneum* **	** *N. hellenicum* **	** *B. dothidea* **	** *E. lata* **	** *L. pistaciae* **
Control	3.80 ± 0.10 a	4.53 ± 0.07 a	4.08 ± 0.01 a	4.84 ± 0.06 a	1.95 ± 0.0 ^ns^
*B. amylo.* QST713	2.40 ± 0.08 b	2.70 ± 0.08 bc	2.30 ± 0.05 b	3.02 ± 0.06 b	1.77 ± 0.10
*B. amylo.* MBI600	2.48 ± 0.12 b	2.83 ± 0.13 b	2.47 ± 0.08 b	3.22 ± 0.16 b	2.00 ± 0.01
*B. amylo.* D747	2.72 ± 0.15 b	1.93 ± 0.32 c	2.55 ± 0.08 b	2.80 ± 0.15 b	1.92 ± 0.07

^y^ Values derived from three replicated Petri dishes with standard error of the mean (SEM). Means followed by different letters within the column are significantly different according to Tukey’s honest significance difference test (*α* = 0.05). ns = not significant data.

**Table 3 pathogens-11-00829-t003:** Analysis of variance for average lesion caused by *Neofusicoccum hellenicum*, *Neofusicoccoum mediterraneum*, *Botryosphaeria dothidea*, *Eutypa lata* and *Leptosillia pistaciae* isolates on pistachio twigs among 6 different biological treatments in 2 trials.

Factors		Average Lesion (cm) on Pistachio Twigs ^y^									
		*N. mediterraneum*		*N. hellenicum*		*B. dothidea*		*E. lata*		*L. pistaciae*	
	Df	F	*p* Value	F	*p* Value	F	*p* Value	F	*p* Value	F	*p* Value
*Treatment*	6	0.987	0.4525 ^ns^	3.372	*0.0125*	0.767	0.6022 ^ns^	1.298	0.2902 ^ns^	3.636	*0.0085*
*Treatment × trial*	6	0.640	0.6970 ^ns^	1.296	0.2914 ^ns^	1.290	0.2936 ^ns^	0.730	0.6290 ^ns^	1.952	0.1069 ^ns^

^y^ df = degrees of freedom, F test of fixed effects, and *p* value associated to F; ns = not significant data.

**Table 4 pathogens-11-00829-t004:** Post-hoc analysis on main effects of biological treatments in reducing twig infections caused by *Neofusicoccum hellenicum, Neofusicoccoum mediterraneum, Botryosphaeria dothidea, Eutypa lata* and *Leptosillia pistaciae* on pistachio twigs.

Treatment	Average Lesion (cm) on Pistachio Twigs				
	*N. mediterraneum*	*N. hellenicum*	*B. dothidea*	*E. lata*	*L.* *pistaciae* ^y^
Control	14.29 ± 0.29 ^ns^	14.46 ± 0.29 a	12.50 ± 0.67 ^ns^	7.75 ± 1.25 ^ns^	1.27 ± 0.15 a
*T. asperellum* T34	14.21 ± 0.87	13.87 ± 0.29 ab	13.83 ± 1.50	5.23 ± 1.97	1.07 ± 0.02 ab
*T. harzianum* T22	14.83 ± 1.00	11.79 ± 1.29 b	11.58 ± 0.25	2.81 ± 0.44	1.13 ± 0.05 ab
*T. atroviride* SC1	12.33 ± 0.33	13.33 ± 1.00 ab	12.46 ± 1.37	2.54 ± 0.54	1.17 ± 0.33 ab
*B. amylo.* QST713	13.58 ± 0.67	12.37 ± 0.62 ab	11.71 ± 0.21	4.07 ± 0.85	0.91 ± 0.29 ab
*B. amylo.* MBI600	13.96 ± 0.04	13.52 ± 0.65 ab	12.08 ± 0.67	5.62 ± 1.12	0.66 ± 0.01 ab
*B. amylo.* D747	13.95 ± 0.08	12.75 ± 0.08 ab	12.00 ± 0.83	4.77 ± 1.73	0.59 ± 0.02 b

^y^ Data averaged from two trials with standard error of the mean (=SEM) value. Means derived from three samples detached from ten young plants. Means followed by different letters within the column are significantly different according to Tukey’s honest significant difference test (*α* = 0.05). ns = not significant data.

**Table 5 pathogens-11-00829-t005:** Analysis of variance for average lesion caused by *Neofusicoccum mediterraneum* P107 on pistachio fruits among 6 biological treatments at two different rates in 2 trials.

Factor(s)	Fruit Decay Caused by *Neofusicoccum mediterraneum* ^y^				
		Lesion Diameter (cm)		Lesion Surface (cm^2^)	
	df	*F*	*p* value	*F*	*p* value
*Treatment*	12	4.200	*0.000134*	8.1796	*0.000000*
*Trial*	1	1.926	0.171124 ^ns^	1.4758	0.229914 ^ns^
*Treatment × trial*	12	0.862	0.588410 ^ns^	1.3170	0.237622 ^ns^

^y^ df = degrees of freedom, F test of fixed effects, and *p* value associated to F; ns = not significant data.

**Table 6 pathogens-11-00829-t006:** Post-hoc analysis on main effects of biological (label and sublabeled rates) and chemical treatments in reducing artificial infections of pistachio fruit by *Neofusicoccum mediterraneum*.

Treatment	Fruit Decay Amount ^y^	
	Mean Diam. (cm)	Mean Area (cm^2^)
Control	1.81 ± 0.17 a	2.53 ± 0.49 a
*Bacillus amyloliquefaciens* MBI600 (low dosage)	1.27 ± 0.03 ab	1.01 ± 0.12 b
*Bacillus amyloliquefaciens* QST713 (low dosage)	1.22 ± 0.04 b	0.94 ± 0.10 b
*Trichoderma harzianum* T22 (low dosage)	1.22 ± 0.05 b	0.97 ± 0.12 b
*Trichoderma harzianum* T34 (low dosage)	1.22 ± 0.04 b	0.98 ± 0.09 b
*Bacillus amyloliquefaciens* D747 (low dosage)	1.17 ± 0.03 b	0.92 ± 0.11 b
*Trichoderma atroviride* SC1 (low dosage)	1.11 ± 0.004 b	0.92 ± 0.01 b
*Bacillus amyloliquefaciens* MBI600 (medium dosage)	1.05 ± 0.08 b	0.78 ± 0.10 b
*Bacillus amyloliquefaciens* QST713 (medium dosage)	1.05 ± 0.01 b	0.70 ± 0.01 b
*Trichoderma atroviride* SC1 (medium dosage)	1.05 ± 0.22 b	0.75 ± 0.34 b
*Trichoderma harzianum* T22 (medium dosage)	0.97 ± 0.02 b	0.60 ± 0.01 b
*Trichoderma harzianum* T34 (medium dosage)	0.92 ± 0.02 b	0.64 ± 0.04 b
*Bacillus amyloliquefaciens* D747 (medium dosage)	0.91 ± 0.25 b	0.77 ± 0.28 b

^y^ Data averaged from two trials and with standard error of the mean (SEM). Means followed by different letters within the column are significantly different according to Tukey’s honest significance difference test (*α* = 0.05).

**Table 7 pathogens-11-00829-t007:** Bioformulates selected for experiments.

Active Ingredient	Trade Name	Rates (g or mL/1000 L)	
		Low	Medium
*Bacillus amyloliquefaciens* QST713 (formerly *B*. *subtilis*)	Serenade^®^ Aso	2700	5300
*Bacillus amyloliquefaciens* ssp. *plantarum* D747	Amylo-X^®^ LC	2000	3500
*Bacillus amyloliquefaciens* MBI600	Serifel^®^	500	1000
*Trichoderma harzianum* T-22	Trianum-P^®^	2500	5000
*Trichoderma asperellum* T34	T34 Biocontrol^®^	10	500
*Trichoderma atroviride* SC1	Vintec^®^	1000	2000

## Data Availability

The data presented in this study are available on request from the corresponding author.

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
