# Peer review of "Potential Role of Biocontrol Agents for Sustainable Management of Fungal Pathogens Causing Canker and Fruit Rot of Pistachio in Italy"

_pathogens, 2022, doi:10.3390/pathogens11080829_

Round 1
Reviewer 1 Report
Dear Authors,
Your publication is sound interesting as I trying to address the Botryosphaeriaceae canker pathogens. However, I need to clarify a few more things. in your experiment did you have both positive and negative control? what is the effect of Trichoderma alone as a pathogen? on these hosts. In the discussion please discuss these facts. In addition mention how the strains were initially obtained? they might be not isolated by yourself yet, in the original study are they isolated as endophytes?
Additional comments are given in the file

Author Response
Point 1: Dear Authors,
Your publication is sound interesting as I trying to address the Botryosphaeriaceae canker pathogens. However, I need to clarify a few more things. in your experiment did you have both positive and negative control? what is the effect of Trichoderma alone as a pathogen? on these hosts. In the discussion please discuss these facts. In addition mention how the strains were initially obtained? they might be not isolated by yourself yet, in the original study are they isolated as endophytes?
Response 1: Dear reviewer, thanks for your comments. Regarding the positive and negative control: in all the experiments as positive control we used the pathogen alone, with no treatments. We did not use negative control (no inoculation and no treatments) since we collected these shoots from a healthy orchard and all the shoots were new green suckers with no external and internal symptoms. If some shoot showed any kind of discolouration it was discarded and not used in the experiments. Regarding the Trichoderma as a pathogen, we already used these bioformulates during field experiments and we did not observe any avverse effect of this Trichoderma on pistachio as well as other crops. The isolates used in this study derived from other studies already published (see the references mentioned next to the isolate description in M&M section). Specifically, these isolates were registered in the CBS collection as well as Genebank. We isolated them as pathogens, some Botryosphaeria spp. occasionally as endophytes, but mostly as pathogens. But in the case of these isolates all of them derived from symptomatic samples and they have been tested in pathogenicity tests.

Reviewer 2 Report
The authors present a manuscript on the potential role of biocontrol agents for sustainable management of fungal pathogens causing canker and fruit rot of pistachio in Mediterranean Italy. The experiments were conducted with commertially labeled biofungicides for agricultural usage. The topic is of relevancy, specially for sustainable farming aiming to reduce pesticides (fungicides) residues on pre- and postharvest in attendance to the European Commission’s “Farm to Fork Strategy”, which aims a reduction of 50% of pesticides usage by 2030. The experimental design is sound for the hypothesis testing on the efficacy of biocontrol agents (BCAs) against the main canker and fruit fungal pathogens in pistachio from Italy, as it included not only in vitro dual culture assays but also in vivo assays with detached twigs and fruits.
My general recommendation is for accepting the manuscript for publication. In general the manuscript is well written. I would like, though, to encorage the authors to submit the manuscript for English editing (As I am not an English native speaker myself) and implement very few written style changes in the manuscript to improve the flow, freadability and understanding of the science presented.
Otherwise my comments are for very minor corrections.
In the Abstract: Please insert a phrase providing information on what pistachio pathogens / diseases were targeted for biological control and the general importance of these pathogens. For example, you cited dieback caused by Cytospora pistaciae, Eutypa lata and Leptosillia pistaciae, fruit and foliar diseases caused by Arthrinium xenocordella and Tuberculina persicina and Septoria pistaciarum as the main fruit/foliar pathogens from pistachio in Italy. You also mentioned fungi such as Botryosphaeria dothidea, Neofusicoccum hellenicum and N. mediterraneum infecting fruit clusters, leaves and shoots. Summarize these pathogens/diseases in the Abstract according to their importance, as convenient. Also, provide species full names for the Trichoderma based biofungicides used.
In the discussion the authors present their data as preliminary. I recommend to get rid of this adjective because the study by itself is not preliminary at all and the data presented is robust enough for deriving relevant information. Perhaps it should be clearly stated that the BCAs tested are commercially labled biofungicides.
On the M&M section: My interpretation is that a pitfall in the methodology of applying / inoculating both the BCAs and the pathogen concomitantely to the detached twigs on very harsh (and large) wounds compromised the biocontrol ability of the selected BCAs. In the M&M section indicated that: i) "Once completely dry, two wounds for each twig were made using a cork borer (6 mm diameter); and ii)"Each twig was immersed for 10 minutes in the BCAs water solution (the lowest label dosage according to the label recommendations) and air dried before applying the mycelial plug (6 mm diameter) of each pathogen onto the wounds." Why have you decided to inoculate simultaneously other than offering an interval of about 5 days of a week before the pathogens´ inoculation? Alternatively, this would give chances for the BCA to colonize the host tissues, increase in numbers, secrete metabolites (with antagonistic properties) and perhaps induce resistance on the plants. Also, why have you chosen the lowest dose instead of the recommended dose or even the highest dose? I suspect that at the lowest dose the BCA were unable to stablish themselves on the twigs to provide any biocontrol against the pathogens. Besides, these BCAs are supposedly not harmful for the environment, even at the highest dosage. Fruits were also treated with the BCAs and inoculated with the fungal pathogens within a very short interval, not leaving a window of time for the biocontrol agents to begin their antagonistic activity.
Discuss this in these aspects of the timing between BCA application and the pathogen´s inoculation the methodology in the Discussion section and reflect upon necessary changes, if you will.
Finally, it would be very important to put into perspective in the discussion section that innitiaves of bioprospection for BCAs derived from pistachio agroecosystems with low disease incidence (perhaps under suppressive natural conditions) could be another venew for development of new biofungicides for sustainable pistachio production, since the ones labeled and available in the market seemed not very efficacious.
Author Response
Response to Reviewer 2 Comments
Minor Comments
Point 1: The authors present a manuscript on the potential role of biocontrol agents for sustainable management of fungal pathogens causing canker and fruit rot of pistachio in Mediterranean Italy. The experiments were conducted with commertially labeled biofungicides for agricultural usage. The topic is of relevancy, specially for sustainable farming aiming to reduce pesticides (fungicides) residues on pre- and postharvest in attendance to the European Commission’s “Farm to Fork Strategy”, which aims a reduction of 50% of pesticides usage by 2030. The experimental design is sound for the hypothesis testing on the efficacy of biocontrol agents (BCAs) against the main canker and fruit fungal pathogens in pistachio from Italy, as it included not only in vitro dual culture assays but also in vivo assays with detached twigs and fruits.
My general recommendation is for accepting the manuscript for publication. In general the manuscript is well written. I would like, though, to encorage the authors to submit the manuscript for English editing (As I am not an English native speaker myself) and implement very few written style changes in the manuscript to improve the flow, freadability and understanding of the science presented.
Response: the manuscript was revised by a collegue affiliated to U.C. Davis and currently working in the USA.
Otherwise my comments are for very minor corrections.
In the Abstract: Please insert a phrase providing information on what pistachio pathogens / diseases were targeted for biological control and the general importance of these pathogens. For example, you cited dieback caused by Cytospora pistaciae, Eutypa lata and Leptosillia pistaciae, fruit and foliar diseases caused by Arthrinium xenocordella and Tuberculina persicina and Septoria pistaciarum as the main fruit/foliar pathogens from pistachio in Italy. You also mentioned fungi such as Botryosphaeria dothidea, Neofusicoccum hellenicum and N. mediterraneum infecting fruit clusters, leaves and shoots. Summarize these pathogens/diseases in the Abstract according to their importance, as convenient. Also, provide species full names for the Trichoderma based biofungicides used.
Response: Ok, thank you for the comment. See revised manuscript (In attachment).
In the discussion the authors present their data as preliminary. I recommend to get rid of this adjective because the study by itself is not preliminary at all and the data presented is robust enough for deriving relevant information. Perhaps it should be clearly stated that the BCAs tested are commercially labled biofungicides.
Response: Thank you for precious suggestion. We changed the term as you recommend.
On the M&M section: My interpretation is that a pitfall in the methodology of applying / inoculating both the BCAs and the pathogen concomitantely to the detached twigs on very harsh (and large) wounds compromised the biocontrol ability of the selected BCAs. In the M&M section indicated that: i) "Once completely dry, two wounds for each twig were made using a cork borer (6 mm diameter); and ii)"Each twig was immersed for 10 minutes in the BCAs water solution (the lowest label dosage according to the label recommendations) and air dried before applying the mycelial plug (6 mm diameter) of each pathogen onto the wounds." Why have you decided to inoculate simultaneously other than offering an interval of about 5 days of a week before the pathogens´ inoculation?
Response: Regarding the effects of wound, we think that this methodology does not affect the biocontrol activity since we try to simulate the pruning wound in field. According to previous experience we noticed that pistachio shoots dry quickly compared to other species. Once dried internally it is impossible to distinguish any kind of canker/wood discolouration caused by the pathogen. For these first trials, the authors did not include preventive and curative applications but unique thesis about simultaneous application.
Alternatively, this would give chances for the BCA to colonize the host tissues, increase in numbers, secrete metabolites (with antagonistic properties) and perhaps induce resistance on the plants. Also, why have you chosen the lowest dose instead of the recommended dose or even the highest dose?
I suspect that at the lowest dose the BCA were unable to stablish themselves on the twigs to provide any biocontrol against the pathogens. Besides, these BCAs are supposedly not harmful for the environment, even at the highest dosage. Fruits were also treated with the BCAs and inoculated with the fungal pathogens within a very short interval, not leaving a window of time for the biocontrol agents to begin their antagonistic activity.
Response: In this study we decided to use the lowest dosage since we focused our attention in seeing if it was possible to observe some results even with the lowest label dosage. These products are expensive for the growers and it would be interesting to see if some results can be reached also with the lowest dosage first before to use the highest dosage (that of course is not harmful for the environment).
The problem of the correct timing of application is still open issue and under investigation in our lab in other experiments. In this case we just waited until the product was perfectly adsorbed by the host tissues before to inoculate the pathogen. Of course each product probably has a perfect timing of application but we decided to follow one rule for all the products, in this case the absorption by the tissues.
Discuss this in these aspects of the timing between BCA application and the pathogen´s inoculation the methodology in the Discussion section and reflect upon necessary changes, if you will.
Finally, it would be very important to put into perspective in the discussion section that innitiaves of bioprospection for BCAs derived from pistachio agroecosystems with low disease incidence (perhaps under suppressive natural conditions) could be another venew for development of new biofungicides for sustainable pistachio production, since the ones labeled and available in the market seemed not very efficacious.
Response: Ok done.

Reviewer 3 Report
The manuscript is well writen and it is addressed to control new pre-and postharvest diseases that cause community concerns. I consider the manuscipt needs minor corrections.
Minor corrections:
1-) Write "in vitro" and species names (Neofusicoccum mediterraneum, Neofusicoccum hellenicum, Botryosphaeria dothidea, Eutypa lata, and Leptosillia pistaciae) in italic style on titles of Table 1 and Table 2 (lines 95, 96, 115, and 116);
2-) Add one line of space between the lines 163 and 164 to separate the Table 5 of the text;
3-) Write "Pistachio and Health Journal" abbreviated - Pistachio and Health J. (Reference 25, line 410).
Modification:
I suggest to reformulate the sentence:
"Based on two assays data, Bacillus amyloliquefaciens D747 provided slightly better effects in reducing mycelial growth of fungal pathogens, if compared with the other Bacillus strains." (lines 121-123)
In general, B. amylo. QST713 (and not the B. amylo. D747) provided slightly better effects in reducing mycelial growth in second assay, because showed higher values of mycelial reduction. While nothing can be concluded from the values shown in the first assay (Table 2).
I think that you can write more specifically about the results of Table 2, as you did in the discussion of Table 1.
Author Response
Response to Reviewer 3 Comments
Point 1: The manuscript is well written and it is addressed to control new pre-and postharvest diseases that cause community concerns. I consider the manuscipt needs minor corrections.
Minor corrections:
1-) Write "in vitro" and species names (Neofusicoccum mediterraneum, Neofusicoccum hellenicum, Botryosphaeria dothidea, Eutypa lata, and Leptosillia pistaciae) in italic style on titles of Table 1 and Table 2 (lines 95, 96, 115, and 116);
Ok, we modified according to editing style of the Journal.
2-) Add one line of space between the lines 163 and 164 to separate the Table 5 of the text;
Done
3-) Write "Pistachio and Health Journal" abbreviated - Pistachio and Health J. (Reference 25, line 410).
Done
Modification:
I suggest to reformulate the sentence:
"Based on two assays data, Bacillus amyloliquefaciens D747 provided slightly better effects in reducing mycelial growth of fungal pathogens, if compared with the other Bacillus strains." (lines 121-123)
Done.
In general, B. amylo. QST713 (and not the B. amylo. D747) provided slightly better effects in reducing mycelial growth in second assay, because showed higher values of mycelial reduction. While nothing can be concluded from the values shown in the first assay (Table 2).
Ok, we rephrased the sentence.
I think that you can write more specifically about the results of Table 2, as you did in the discussion of Table 1.
Response: Ok, done.

Round 2
Reviewer 1 Report
Dear Authors,
Thank you very much for the revised version. few minor corrections.

Author Response
Response to Reviewer 1 Comments
Point 1: Dear Authors. Thank you very much for the revised version. few minor corrections
Response 1: Thank you again for additional suggestions/corrections. Suggested modifications done and other typos in the refernces corrected in the revised MS.
